# Microwave Irradiation and Glutamic Acid-Assisted Phytotreatment of Tannery and Surgical Industrial Wastewater by Sorghum

**DOI:** 10.3390/molecules27134004

**Published:** 2022-06-22

**Authors:** Mujahid Farid, Muhammad Abubakar, Zaki Ul Zaman Asam, Wajiha Sarfraz, Mohsin Abbas, Muhammad Bilal Shakoor, Shafaqat Ali, Sajid Rashid Ahmad, Asim Jilani, Javed Iqbal, Abdullah G. Al-Sehemi, Omar A. Al-Hartomy

**Affiliations:** 1Department of Environmental Sciences, Hafiz Hayat Campus, University of Gujrat, Gujrat 50700, Pakistan; muhammadabubakar2128@gmail.com (M.A.); zaki.asam@uog.edu.pk (Z.U.Z.A.); mohsin.abbas@uog.edu.pk (M.A.); 2Department of Botany, Government College Women University, Sialkot 51310, Pakistan; wajiha.sarfraz@gcwus.edu.pk; 3College of Earth & Environmental Sciences, University of the Punjab, Lahore 54590, Pakistan; bilalshakoor88@gmail.com (M.B.S.); sajidpu@yahoo.com (S.R.A.); 4Department of Environmental Sciences, Government College University, Faisalabad 38000, Pakistan; 5Department of Biological Sciences and Technology, China Medical University, Taichung 40402, Taiwan; 6Centre of Nanotechnology, King Abdul-Aziz University, Jeddah 21589, Saudi Arabia; iqbaljavedch@gmail.com; 7Research Center for Advanced Materials Science (RCAMS), King Khalid University, Abha 61413, Saudi Arabia; agsehemi@kku.edu.sa; 8Department of Chemistry, College of Science, King Khalid University, Abha 61413, Saudi Arabia; 9Department of Physics, Faculty of Science, King Abdulaziz University, Jeddah 21589, Saudi Arabia; oalhartomy@kau.edu.sa

**Keywords:** microwave radiation, glutamic acid, oxidative stress, phytoremediator, sorghum, textile and surgical wastewater

## Abstract

We investigated how different doses of microwave irradiation (MR) affect seed germination in Sorghum, including the level of remediation against textile and surgical wastewater (WW) by modulating biochemical and morpho-physiological mechanisms under glutamic acid (GA) application. The experiment was conducted to determine the impact of foliar-applied GA on Sorghum under wastewater conditions. Plants were treated with or without microwave irradiation (30 s, 2.45 GHz), GA (5 and 10 mM), and wastewater (0, 25, 50, and 100). Growth and photosynthetic pigments were significantly decreased in plants only treated with various concentrations of WW. GA significantly improved the plant growth characteristics both in MR-treated and -untreated plants compared with respective controls. HMs stress increased electrolyte leakage (EL), hydrogen peroxide (H_2_O_2_), and malondialdehyde (MDA) content; however, the GA chelation significantly improved the antioxidant enzymes activities such as ascorbate oxidase (APX), superoxide dismutase (SOD), peroxidase (POD), and catalase (CAT) both in MR-treated and -untreated plants under WW stress compared with respective controls. The results suggested that the MR-treated plants accumulate higher levels of HMs under GA addition in comparison to the WW-only-treated and MR-untreated plants. The maximum increase in Cd accumulation was observed in the range of 14–629% in the roots, 15–2964% in the stems, and 26–4020% in the leaves; the accumulation of Cu was 18–2757% in the roots, 15–4506% in the stems, and 23–4605% in the leaves; and the accumulation of Pb was 13–4122% in the roots, 21–3588% in the stems, and 21–4990% in the leaves under 10 mM GA and MR-treated plants. These findings confirmed that MR-treated sorghum plants had a higher capacity for HMs uptake under GA and could be used as a potential candidate for wastewater treatment.

## 1. Introduction

Overpopulation, increased industrialization, and urbanization have contributed to the emergence of heavy metals (HMs) concentrations in recent decades [1]. Heavy metals stress is one of the most pertinent abiotic factors that has attained wide acceptance due to its long-term repercussions. Textile industries have significantly contaminated soil and water in various areas of Pakistan, especially central Punjab, including the Sialkot district, and have extensively released prolonged toxins such as HMs [2,3]. The textile and surgical industry in Sialkot is rapidly expanding, contributing to the country’s economy [4,5]. Sialkot’s agricultural land is used for conventional farming, and despite the scarcity of fresh water, agricultural practices explicitly use 70% of the produced industries’ wastewater for irrigation purposes after primary treatment [6]. Hazardous dyes, pigments, oil, surfactants, HMs, sulphates (SO_4_^2−^), and chlorides (Cl^−^) have been reported in wastewater of the textile industry. The level of HMs in arable land seems not consistently high, but when treated with wastewater, the HMs content increases. All of these contaminants are potentially damaging to the water quality [7]. Consequently, it has a substantial negative impact on all life forms [8]. Toxins from untreated wastewater are identified as one of the most noticeable polluters of our environment, including water bodies and soils. These toxins are carcinogenic, mutagenic, and genotoxic to all living things [9].

Heavy metals have a significant impact on plant growth. Their toxicity is linked to reactive oxygen species (ROS) generation, which eventually causes redox imbalance [10]. Several environmental factors influence plant growth and development, while heavy metals contamination has become a massive ecological threat [11]. Metals pose a risk to both the environment and biosystems due to their expansion and high conductivity. These elements readily give up their electrons to form cations via ions [12]. Plant and human health are jeopardized due to its transport and sequestration into the food chain from lower to higher trophic levels [13,14]. Heavy metals from the surgical sector can cause environmental problems through multiple mechanisms [15]. The emission of industrial waste from HMs in waterways has devastating effects on the environment’s flora and fauna [16]. Surgical and textile industrial effluents contain nickel (Ni), lead (Pb), cadmium (Cd), copper (Cu), iron (Fe), magnesium (Mn), and chromium (Cr) in various concentrations (Reddy and Osborne, 2020). Some metals (Cu, Ni) are involved in cellular metabolism, while others are not (Cd, Pb). Pb, Cd, and Cu toxicities combine to cause a wide range of oxidative stress [1]. Cd is the most toxic element to humans, animals, and plants due to its high mobility and bioaccumulative capacity and acts as a potential environmental pollutant [17,18]. Cd interferes with several aspects of plant growth, including photosynthesis, nutrient absorption, leaf chlorosis, and assimilation, which causes changes in the structure of chloroplasts, mitochondria, and cell wall components, eventually leading to a reduction in crop yields [19,20]. Even though copper is required for the catalysis of enzymatic biochemical activities in plants, an excess of Cu can inhibit or hinder plant development and promote oxidative stress [21,22]. Lead (Pb) has deleterious morphological, physiological, and biochemical effects on living things. It potentiates plant growth by inhibiting seedling development, root elongation, transpiration, chlorophyll production, cell division, and subsequent plant growth [23,24].

Untreated water is corrosive to the environment and other natural organisms. With these environmental challenges, the toxicity of metals in industrial effluent has attracted the interest of research [25]. All types of wastewater must be treated before dumping into open water to reduce water pollution. Consequently, an environmentally sustainable method of treating textile and surgical wastewater must be developed and implemented. Moreover, phytoremediation is the most acceptable, cost-effective, and environmentally friendly technique [26]. The most appropriate method for treating industrial wastewater is phytoremediation-constructed wetland technology. Plants take up pollutants from water, soil, and air. Eventually, plant species have a spectrum of potential for nutrient and pollutant removal and represent an adequate phytoremediation and stress resistance. Sorghum is among the flowering plants known to remove heavy metals from soil. Sorghum, a member of the Poaceae family, is also known as Great millet and is a valuable tropical and subtropical crop [27,28]. It is also resistant to drought, heat, and salt. Sorghum is extensively cultivated for its high climate versatility, higher yield potential, and improved quality. Even in the presence of heavy metals, sorghum plants were highly resistant to metal pollution and able to produce high biomass [17]. Along with plant applications, numerous eco-friendly wastewater treatment mechanisms, such as plant seeds treated with microwave radiation and amino chelators, are now being used. These strategies and chemical agents have been researched to improve the effectiveness of phytoremediation.

Amino acids (AAs) are imperative for crop development and can be used to mitigate abiotic stress [29]. Within plant cells, amino acids are one of the most influential metabolites. The presence of amino acids influences the physical and chemical properties of plant cells, tissues, and organs. As a result, amino acids associate with remediator plants to boost metal toxicity absorption from polluted environments [30]. Glutamic acid (GA) is an essential amino acid in plant physiology, where it plays pivotal roles in a variety of metabolic processes, including nitrogen assimilation pathways [31]. GA applied at the seedling stage improves plant yield attributes such as biomass, lipids, soluble protein levels, and starch. It also assisted in the plant’s natural process of strengthening [32,33]. Multiple amino acids, primarily glycine, acetic acid, and lysine, have recently been supplemented into nutrient fertilizer formulations in the context of chelate or a simple complex to improve the plant use efficiency of applied fertilizer [34,35]. Exogenous amino acid implementation has proven to endorse agricultural yields in various crops. The foliar application of amino acids has a phenomenal impact under environmental stresses such as salinity or water stress [29,36,37,38].

Another approach is that electromagnetism is considered for biochemical processes and organisms to function. The application of microwave technology to extract plants’ bioactive components is still under-discussed. Moreover, technology is gaining notoriety in industries, primarily the agri-food sector [39]. Electromagnetic waves could have long-term health consequences. Microwave radiation (MR) has been used in a wide range of microbiology and soil science techniques [40]. MR affects plant growth, development, and seed germination. Low-intensity microwaves did not affect plant growth, but increased microwave irradiation has slowed the seed germination rate [41].

However, no research has been conducted on the impact of microwave radiation on the seed of Sorghum plants, using the foliar application of glutamic acid against textile and surgical wastewater. Therefore, the following are the research objectives: (a) to evaluate the effect of microwave irradiation on seed germination; (b) to assess the physiological and biochemical response of Sorghum to wastewater stress; (c) to investigate Sorghum’s phytoremediation potential for heavy metals under glutamic acid and microwave irradiation applications; (d) to study the influence of GA and MR on Sorghum plant resistance to oxidative stress.

## 2. Results

### 2.1. Seed Germination Potential under Microwave Radiation

According to the findings, increased frequency over a longer time period inhibited seed germination. The optimum growth in seed germination was observed at a frequency of 2.45 GHz for 30 s, but at the same frequency for the other time scales of 90 s and 120 s, there was a decrease in seed germination. The maximum germination of seeds was observed at 30 s (9.33) followed by 0 s (7.3), 60 s (6.3), 90 s (5), and 120 s (4.3). The intensity of microwave radiation varies with frequency and duration of exposure, and it has a substantial impact on seed germination potential. The accurate results were obtained with MR (30 s) and were used within the research.

### 2.2. Morphological Characteristics

Different growth parameters of Sorghum were evaluated in the presence of surgical and textile wastewater. Measured growth parameters were plant height, root length, number of leaves, and leaf area. Tested plants were grown in wastewater concentrations (0, 25, 50, 75, and 100%) in the presence and absence of GA. The addition of GA (5 mM and 10 mM) increased growth characteristics, but MR for 30 s led to substantial improvements in the growth characteristics of sorghum plants (Appendix A). A reduction was observed in MR-untreated plants due to heavy metals stress as compared with the control of 31.18% and 28.68% for plant height, 55.97% and 40.51% for root length, 29.70% and 27.85% for leaf area, and 60% and 33.33% for the number of leaves per plant, respectively. Adding GA (5 mM and 10 mM) and MR (the 30 s) enhanced the plant height by 5–43%, root length by 6–96%, leaf area by 12–60%, and number of leaves by 10–175% at different levels of wastewater concentration. The fresh and dry biomass of plants decreased with as the wastewater concentration (Appendix A). Sorghum is a well-known plant for up taking HMs, but the increased concentration of wastewater affects the biomass and growth parameters of plants. Without MR treatment and GA, compared to the control, there was a reduction in the fresh weight of leaves by 64.75% and 53%, stems by 7.57% and 37.32%, and roots by 74.56% and 58.14%, respectively. The maximum reduction in dry biomass was 73.04% and 54.55% for leaves, 40.20% and 26.76% for stems, and 62.15% and 52.44% for roots, respectively. The combination of GA (5 mM and 10 mM) and MR (30 s) increased the fresh weight of leaves by 8–91%, stems by 5–64%, and roots by 10–174%. Similarly, the increase in dry biomass was 11–168% for leaves, 7–75% for stems, and 9–105% for roots.

### 2.3. Biochemical Characteristics

Chlorophyll and carotenoid contents were estimated to assess the photosynthetic reduction caused by the presence of toxins in wastewater. The maximum reduction was observed in MR-untreated plants and the absence of GA in tested plants as compared to the control (Figure 1). Thus, the biochemical characteristics was as follows: chlorophyll a—68.44% and 57.67%, chlorophyll b—70.12% and 53.73%, total chlorophylls—70.24 and 56.96%, and carotenoids—80% and 63.33%, respectively. With GA (5 mM and 10 mM) and MR (30 s) added, it enhanced the content of chlorophyll a by 6–99%, chlorophyll b by 8–156%, total chlorophylls by 8–115%, and carotenoids by 10–188% at different concentrations of WW (0, 25, 50, and 100%).

### 2.4. Soluble Protein and SPAD Value

The soluble protein of sorghum plants was reduced with the increase in the concentration of wastewater, as shown in Figure 2. The maximum reduction for soluble protein due to metals stress as compared to the control was 58.46% and 50.91% in leaves, 72.22% and 60.72% in roots, and 79.77% and 73.22% for SPAD, respectively. With the GA (5 mM and 10 mM) and MR (30 s) added, it enhanced the soluble protein values in leaves by 17–103%, in roots by 14–148%, and for SPAD by 19–137% at various concentrations of wastewater.

### 2.5. ROS Content in Sorghum

Enhanced oxidative stress and electrolyte leakage were estimated in the leaves and roots of MR-untreated plants with increasing heavy metals concentrations (0, 25, 50, 75, and 100%). Heavy metals stress significantly enhanced the production of electrolyte leakage and oxidative stress, particularly MDA and H_2_O_2_ (Figure 3). The maximum increase observed under HMs stress compared to the controls for EL was 385.47% and 1158.06% in roots and 385.06% and 895.93% in leaves; for H_2_O_2_, it was 81.33% and 193.81% in roots and 116.61% and 231.45% in leaves; and for MDA, it was 130.54% and 176.47% in roots and 104.91% and 124.54% in leaves, respectively. The use of GA (5 mM and 10 mM) and MR (30 s) with varying concentrations of WW significantly reduced the ROS content in plants. It decreased the EL content in roots by 5–73% and in leaves by 9–71%, H_2_O_2_ in roots by 8–65% and in leaves by 8–55%, and MDA in roots by 3–66% and in leaves by 4–46%.

### 2.6. Response of Antioxidants in Sorghum

To reduce the stress induced by heavy metals in Sorghum plants, the production of antioxidant enzymes SOD, POD, APX, and CAT was measured in the roots and leaves of sorghum plants grown at different levels of WW concentrations (Figure 4). The maximum increase observed under HMs stress compared to the controls for APX was 1059.86% and 427.09% in leaves and 171.16% and 116.44% in roots; for CAT, it was 1394.85% and 424.92% in leaves and 639.25% and 408.53% in roots; for POD, it was 2198.02% and 472.89% in leaves and 1005.61% and 377.79% in roots; and for SOD, it was 1131.75% and 479.97% in leaves and 766.27% and 398.27% in roots, respectively. With the application of GA (5 mM and 10 mM) and MR (30 s), it reduced the content of APX in leaves by 3–170% and in roots by 4–102%, of CAT in leaves by 2–232% and in roots by 3–99%, of POD in leaves by 3–372% and in roots by 3–170%, and of SOD in leaves by 3–155% and in roots by 2–104%.

### 2.7. Heavy Metals Concentration and Accumulation in Sorghum

The concentration of various metals in Sorghum, including Cd, Pb, and Cu, enhanced significantly with an increase in wastewater concentration. The results showed that using GA and MR alone or in combination increased the Cd, Cu, and Pb uptake and accumulation in plants (Table 1, Table 2 and Table 3). The concentration of Cd increased by 599.33% and 241.15% in roots, 755.83% and 214.49% in stems, and 656.52% and 248.49% in leaves. The concentration of Cu increased by 429.19% and 258.09% in roots, 415.45% and 283.14% in stems, and 488.85% and 303.20% in leaves. The concentration of Pb increased by 634.12% and 245.58% in roots, 757.71% and 259.80% in stems, and 384.98% and 271.31% in leaves. The accumulation of Cd increased by 203.11% and 94.57% in roots, 475.80% and 149.26% in stems, and 155.44% and 89.46% in leaves. The accumulation of Cu increased by 129.49% and 103.59% in roots, 246.53% and 203.57% in stems, and 99.47% and 119.25% in leaves. The accumulation of Pb increased by 217.53% and 96.65% in roots, 477.40% and 185.19% in stems, and 64.17% and 101.85% in leaves, respectively. Correspondingly, the application of GA (5 mM and 10 mM) and MR (30 s) under wastewater significantly enhanced the concentration and accumulation in plants as compared to wastewater treated-only plants.

The application of GA (5 mM and 10 mM) and MR (30 s) increased the concentration of Cd by 3–350% in roots, 3–2053% in stems, and 3–2445% in leaves. The concentration of Cu increased by 3–1685% in roots, 3–3150% in stems, and 4–2900% in leaves. The concentration of Pb increased by 2–2445% in roots, 4–2440% in stems, and 4–3150% in leaves. The accumulation of Cd increased by 14–629% in roots, 15–2964% in stems, and 26–4020% in leaves. The accumulation of Cu increased by 18–2757% in roots, 15–4506% in stems, and 23–4605% in leaves. The accumulation of Pb increased by 13–4122% in roots, 21–3588% in stems, and 21–4990% in leaves at different levels of WW, respectively.

## 3. Discussion

Plants take macro-elements as nutrient content on a minuscule scale. Increased levels of these metals have physiological, morphological, and metabolic effects on organisms, and they may impact human life as they enter the food chain [42]. Cd, Cu, and Pb are toxic compounds with no discernible role in plant metabolism. The excessive use of industrial effluents alters soil properties and has severe implications for food safety. Specific management strategies should be used under these conditions to improve plant growth and productivity [43,44]. Plant resistance to HM stress is dependent on an intricate system of molecular and physiological mechanisms. In the presence of a robust antioxidant defense system, these plant mechanisms make plant species hyper-tolerant to HMs stress [45]. Compared to control values, increasing wastewater concentrations (25, 50, 75, and 100%) significantly reduces plant morphological characteristics (Appendix A). The findings are similar to those reported by Hassan et al. [46], who found that HMs reduce the physiochemical activities of sorghum. Plant physiological processes suppress, having caused morphological changes. Furthermore, the chelating agent GA markedly increased *Brassica napus* growth performance when subjected to metal stress [47]. Plants treated with microwave irradiation respond positively to glutamic acid application. Subsequently, MR and GA demonstrate a survival benefit in Sorghum plants against the presence of toxic substances in wastewater. Microwave interactions in germination can be attributed to the microwave’s thermal effects, and thus low humidity content can be likened to the grain’s initial moisture content [48]. Sorghum plants are well-positioned to contribute to resisting HMs when regarded with MR and GA.

Plants that were not subjected to wastewater, treated only with the chelating agents GA (10 mM) and MR (30 s), had the highest chlorophyll content. Microwave irradiation promotes seed germination when used at low levels [49]. As the concentration of wastewater effluents (25, 50, 75, and 100%) increased, so did the plant’s chlorophyll and carotenoid content (Figure 1). Our findings back up previous research by Zhang et al. [50], who observed that Cd and Zn stress reduced the chlorophyll value of tobacco leaves. The utilization of wastewater generally resulted in changes in the physicochemical properties of the soil and, as a result, increased heavy metals uptake by plants [51]. The increase in chlorophyll content is specifically correlated to plant food production. These chlorophyll pigments actively engage in photosynthetic processes [13,52]. Nonetheless, this study reveals that microwave irradiation at the initial phases of seed germination and the exogenous application of glutamic acid on Sorghum plants tend to increase photosynthetic activity, which increases yield production. Following microwave irradiation with low doses, the chlorophyll content of barley increased [41]. The electrons of hydrogen and magnesium atoms absorb the energy of microwave radiation in chlorophyll molecules [53]. The absorbed energy is split and changes the chlorophyll molecule. If microwave radiation accelerates seed potato germination, subsequent plant ontogenesis levels such as vegetation cycle reduction or cropping size will be influenced [54]. Other crop plants use the foliar amino acid application to boost biomass and photosynthetic activity [55,56]. Glutamic acid is also bolstered in the protection and preservation of yield production and also biological characteristics in the environment [7]. Chelating agents such as lysine, glycine, and glutamic acid assist in the retention of plant yield and photosynthetic pigments [29,57,58].

The soluble protein and SPAD value decrease in conjunction with the metal toxins present in surgical and textile effluents, as shown in the results (Figure 2). According to Farid et al. [59], the lowering in SPAD value could be due to heavy metals stress in the plant. Pb, Cd, Cu, and Zn stress diminished plant soluble protein and SPAD changes at different levels of WW (without GA and MR treatment), as seen in Figure 2. Heavy metals stress increases, as does the production of ROS, causing a reduction in soluble protein [60]. As previously discussed, the value of SPAD tends to decrease due to heavy metals stress, which is consistent with the current research findings [61,62]. Soil application of amino acids substantially increased sweet basil growth, including plant height and the leaf SPAD value of plant leaf extract, compared to unfertilized control plants [63]. Because of the reduction in ROS and EL production, the application of organic acid as a chelator raises the SPAD value and even the soluble protein value [64,65].

The antioxidant properties of four enzymes, POD, APX, SOD, and CAT, are crucial for resistance mechanisms against ROS [66]. Multiple levels of WW (without GA and MR) had the lowest antioxidant concentrations. Surprisingly, the exogenous application of 5 mM and 10 mM GA to MR-treated Sorghum-tested plants elevated antioxidant levels (Figure 4). Foliar amino acid supplementation is an advanced way of improving plant growth, yield, and a range of quality attributes, along with antioxidant activities, especially under adverse environmental conditions [67,68]. The elevated levels of SOD, POD, APX, and CAT disrupt the chemical nature of the soil. It may also have an impact on metabolism and growth [69]. POD enzymes convert H_2_O_2_ to H_2_O and O_2_ in plant cells, SOD and APX catalyze the dissociation of O_2_ into molecular O_2_ and H_2_O_2_, and CAT acts directly on the removal of H_2_O_2_ generated by fatty acid β–oxidation [70]. When plants are stressed, the activity of these enzymes is regularly accelerated, scavenging surplus free radicals and preserving free radicals at physiological dynamic levels to promote resistance mechanisms. Free amino acids perform various functions and have benefits in plant cells, including semi-hormonal effects, osmotic adjustment, and frequently acting as an antioxidant for cell components, including the membrane surface [71,72]. GA application culminated in a relative increase in POD, APX, SOD, and CAT activities in sorghum plants under water stress conditions. Therefore, GA can dramatically improve POD, APX, SOD, and CAT activities under HMs stress, enhancing plants’ ability to scavenge ROS and, inevitably, reducing MDA-induced damage.

Toxins in wastewater accelerated electrolyte leakage and the production of reactive oxygen species. Heavy metals in effluents reduced the ROS content, particularly malondialdehyde (MDA) and H_2_O_2_, according to this research, and the findings are consistent with [73,74,75]. However, with disruptions in the flow of the K+ and electron transport chains, the levels of EL, O_2_^−^, OH, and H_2_O_2_ are rising. It also indicated that increasing the concentration of glutamic acid reduces the oxidative stress caused by MDA under HM stress. In the present research, microwave irradiation-treated plants showed decreased ion leakage, which related to exposure doses having positive effects, consistent with Demidchik et al. [76]. Moreover, the use of amino acids as a chelator assisted the plant in repairing the damage while also enhancing gas exchange and antioxidant properties [77,78]. The antioxidant activities of the leaf samples were reduced when they were applied to multiple concentrations of wastewater effluents. Plant growth suffers the consequences of metal toxicity due to ROS production [79]. The antioxidative mechanism in plants is a natural protective mechanism that intervenes in the disruption induced by oxidative stress. Plants accumulate HMs, which also cause cell damage, a decline in enzymatic activity, and alter plant growth under stress conditions [80,81]. Metal toxicological stress increased MDA content in rape seedlings, indicating plant cytoplasmic membrane peroxidation [82]. HMs stress could severely reduce oxygen utilization, associated with excess ROS such as O_2_ and H_2_O_2_, causing cell membrane damage and influencing crop production. GA, a chelating agent, enhances plant recovery by modifying gas exchange characteristic traits and the antioxidant mechanism. It improves plant growth parameters as well [33].

In the present research, increasing the concentration of wastewater led to substantial metal accumulation in all parts of the plant (Table 1, Table 2 and Table 3). The collection of HMs in plants had toxic effects on plant physiological activities, halting plant growth [83]. In the long term, the heavy metals content of irrigation waters could lead to some accumulations. In this way, soil pH would be a corrosion inhibition precaution against metal risks [84]. The current study was designed to remove as many HMs from wastewater as possible for environmental remediation purposes. The addition of GA improved plant growth against HMs stress, while GA (5 and 10 mM) and MR (the 30 s) collectively enhanced the maximum level of concentration and accumulation. Furthermore, GA strengthens plants’ potential to withstand and tolerate higher levels of HMs [69]. These findings demonstrated that the exogenous application of glutamic acid at moderate concentrations optimizes sorghum growth, specifically leaf greenness and photosynthetic traits, and overcomes oxidative stress. This accomplishment illustrates that the amino acid approach and seeds treated with microwave irradiation accelerated the remediation potential of sorghum, making it the best candidate for phytoremediator. Therefore, chemical fertilizers often neglect several adverse effects on soil physicochemical characteristics. Furthermore, the foliar amino acid application can boost the amino acid content of plant tissues, improving food quality and nutritional value [35]. The study also discussed the effect of microwave irradiation on plants. Because it is a well-known issue, much additional research has been conducted. We theorized that they do impact it, but it was still a challenging issue to explain, so it was noticed that lower doses affected seed germination. Amino acids could even generate multiple complexes with metals and move more freely across the plant [85]. Even so, Sorghum exists naturally and capably in wastewater treatment, but the addition of GA increases the remediation potential.

## 4. Materials and Methods

### 4.1. Growth Conditions and Experimental Setup

Sorghum plants grown hydroponically have been studied for their capacity to absorb and tolerate Cd, Pb, and Cu concentrations. The seeds were taken from the Ayub Agriculture Research Institute (AARI) in Faisalabad, whereas textile and surgical wastewater were collected from a textile and surgical industry on Sialkot’s Wazirabad Road. The surface of the seeds was disinfected by spraying them with 3% H_2_O_2_ for 5 min and then washing them with deionized water. After drying, they were placed in a microwave oven with a frequency of 2.45 GHz for 0, 30, 60, 90, and 120 s time periods. Furthermore, the 10 seeds treated with microwave radiation for each selected time period were placed in petri dishes containing wet paper and incubated in a thermostat incubator for two weeks to observe the germination rate. Optimum germination was observed at 30 s of exposure to a frequency of 2.45 GHz. The maximum germination (mean of three replicates) of seeds was observed at 30 s (9.33) followed by 0 s (7.3), 60 s (6.3), 90 s (5), and 120 s (4.3). The highest germination time period (30 s) was selected for further experiment along with controls (0 s). The untreated and 30 s treated seeds were planted in the soil for 4 weeks. After that, the juvenile plants were washed with deionized water and shifted to hydroponic conditions. The Hoagland nutrient solution was provided in a hydroponic system composed of 5 L glass containers. Roots were immersed in the liquid solution while the shoots were raised above the culture medium. The plants were kept aerated at all times by air pumps. Two weeks after being transplanted, the plants were subjected to various treatments. Treatments were planned at varying concentrations of glutamic acid (5 and 10 mM), microwave radiation (0 and 30 s), and combined textile and surgical wastewater (WW) (25, 50, 75, and 100%), as given in Appendix A. Appendix A shows the physicochemical properties of wastewater (WW) measured using standard procedures, including biological oxygen demand (BOD), electrical conductivity (EC), chemical oxygen demand (COD), and also pH and HMs of the selected textile and surgical wastewater. The plants were maintained at 25 °C with a 16 h photoperiod in a completely randomized design experiment (CRD). The pH of the nutrient was maintained at 5.8–6.3 throughout the activity by adding 1 M of H_2_SO_4_. Controls and treatments were performed in triplicate for analysis.

### 4.2. Analysis of Morpho-Physiological Response in Plants

After four weeks of treatment, the plants were harvested for further evaluation. An electric weighing balance was used to measure the fresh biomass of the Sorghum plant. Sorghum plant material was dried in an oven at 90 °C for 72 h, and dry biomass was obtained. The measuring ruler was used to measure the length of the tested plants. Leaf area was also calculated by using the equation described by Sarfraz et al. [13]. The chlorophyll a, b, and carotenoid content of fresh leaves (0.2 g) were measured using a spectrophotometer described by Metzner et al. [86].

### 4.3. Determination of Antioxidant Enzymes, SPAD Value, and Soluble Protein Content

Sorghum plants’ antioxidant enzyme activities were measured by using different protocols. The method proposed by Nakano and Asada [87] and Aebi [88] was used for estimating the concentrations of ascorbate oxidase (APX) and catalase (CAT). Zhang [89] described a method for calculating superoxide dismutase (SOD) and peroxidases (POD). To determine the concentration of soluble proteins in the *Sorghum* plants, Bradford [90] recommended using Coomassie brilliant blue G-250 as a dye and albumin as a standard. A SPAD-502 (Zhejiang Top Instruments Co., Ltd., Hangzhou, China) meter was used to determine the soil-plant analysis development (SPAD).

### 4.4. Evaluation of Electrolyte Leakage (EL), Malondialdehyde (MDA), and Hydrogen Peroxide H_2_O_2_

Dionisio-Sese and Tobita [91] described the following protocol to measure electrolyte leakage. The leaves were cut into small chunks and placed in the test tube with 8 mL of distilled water. The sample was autoclaved at 121 °C for 20 min after being immersed in water for 2 h to determine the electrical conductivity (*EC*_1_) at the initial level. To assess the final electrical conductivity (*EC*_2_), we should first cool the sample to 25 °C. The electrolyte leakage was calculated using a conductivity meter (model 720, INCO-LAB Company, Kuwait) or pH, and the formula is as follows:*EL* = (*EC*_1_/*EC*_2_) × 100

The contents of hydrogen peroxide and malondialdehyde in plant roots and leaves were measured using the methods described by Heath and Pascker [92], with some modifications suggested by Dhindsa et al. [93] and Zhang and Kirham [94].

### 4.5. Assessment of Heavy Metals Concentration

The 0.5 g sorghum plant was harvested and dried in an electric furnace at 650 °C before being ground into powder. To achieve optimal digestion, concentrated sulfuric acid (H_2_SO_4_) and hydrochloric acids (HCl) were added to the ash at 250–300 °C, followed by the addition of H_2_O_2_ drop by drop. Following digestion, the sample was filtered through (Whatman No. 1) filter paper, and the solution was diluted to 100 mL with distilled water. For further sample analysis, an atomic absorption spectrophotometer (NOVA A400, Analytik Jena, Germany) was used to evaluate the Cu, Pb, and Cd content using the equation proposed by Farid et al. [95].

### 4.6. Statistical Analysis

The statistical analysis was performed on collected data by using Statistix 10.0 to find out the significant difference in/among applied treatments. The data presented in this study are the average value of three replicates for each treatment. ANOVA followed by Tukey’s post hoc test was accomplished to estimate the significant difference at *p* > 0.05.

## 5. Conclusions

The current study focused on the impact of microwave irradiation on the germination of seeds of Sorghum. Although seeds were exposed to microwave radiation (MR) frequencies at different times, the increased germination rate was observed in a relatively short period. Plants responded positively to relatively short microwave irradiation. In addition, it revealed that treated plants were more resistant to heavy metals (HMs) stress than MR-untreated plants were. The maximum accumulation of heavy metals cadmium (Cd), copper (Cu), and lead (Pb) was also observed in MR-treated plants with the addition of glutamic acid (GA, 5 mM, and 10 mM). Chelating agents mixed with microwave irradiation improved maximum growth parameters and photosynthetic characteristics. Plant physiology and biochemical processes were also affected in plants treated only with different levels of wastewater (25, 50, 75, and 100%). The plants treated with 100% wastewater showed the highest reduction in overall growth in all parts of the plant. The chelating agents GA and MR, alone or in combination, increased heavy metals concentrations and accumulation in Sorghum. The combination of GA and MR had the most optimistic implications of a high growth rate and prominent metal accumulation. The research found that controlling doses and minimizing seed exposure time to microwave irradiance have a beneficial influence on plants. Furthermore, it improves the understanding of the GA-dependent role in sorghum plants; however, extensive fieldwork is still necessary to determine the interaction of metals with GA and MR treatment.

## Figures and Tables

**Figure 1 molecules-27-04004-f001:**
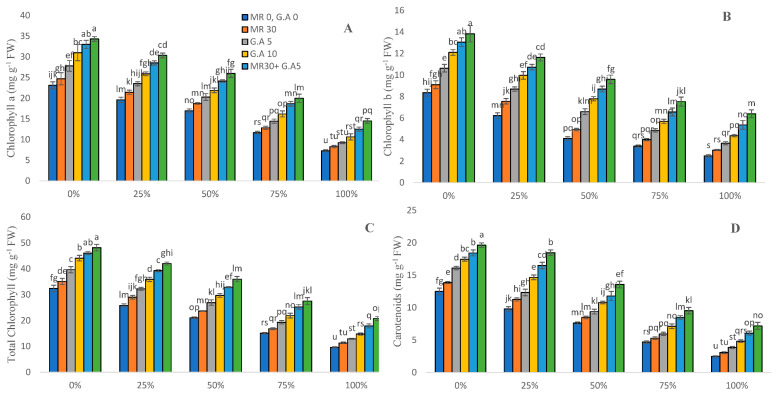
Surgical and textile effluent and glutamic acid effect on Chl a (**A**), Chl b (**B**), total Chl (**C**), and carotenoids (**D**) in *Sorghum*. Values are demonstrated as means of three replicates along with standard deviation. Different small letters indicate that values are significantly different at *p* < 0.05.

**Figure 2 molecules-27-04004-f002:**
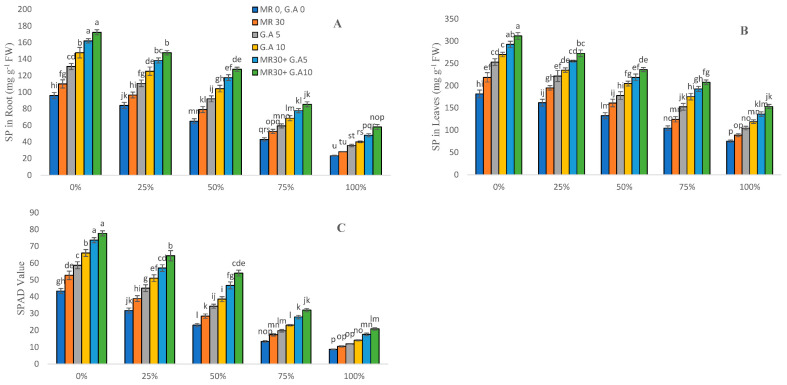
Surgical and textile effluent and glutamic acid effect on SP in root (**A**), SP in leaves (**B**), and SPAD value (**C**) in *Sorghum* L. Values are demonstrated as means of three replicates along with standard deviation. Different small letters indicate that values are significantly different at *p* < 0.05.

**Figure 3 molecules-27-04004-f003:**
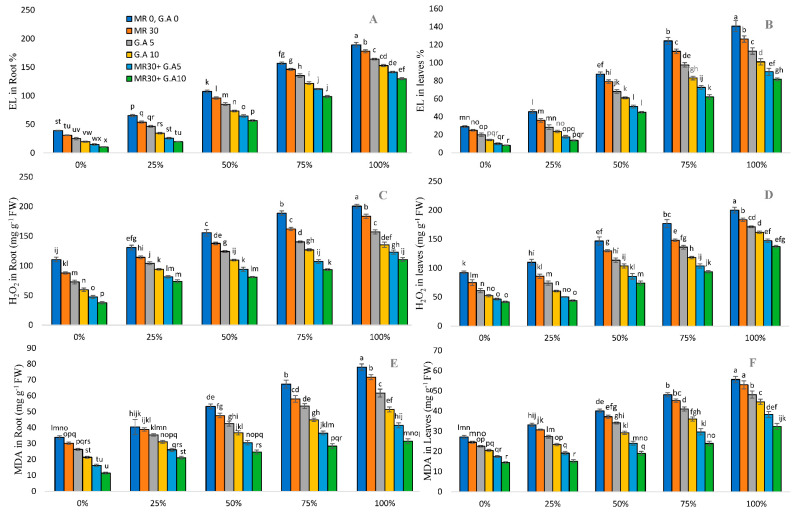
Surgical and textile effluent and glutamic acid effect on EL in root (**A**), EL in leaves (**B**), H_2_O_2_ in roots (**C**), H_2_O_2_ in leaves (**D**), MDA in roots (**E**), and MDA in leaves (**F**) in *Sorghum* L. Values are demonstrated as means of three replicates along with standard deviation. Different small letters indicate that values are significantly different at *p* < 0.05.

**Figure 4 molecules-27-04004-f004:**
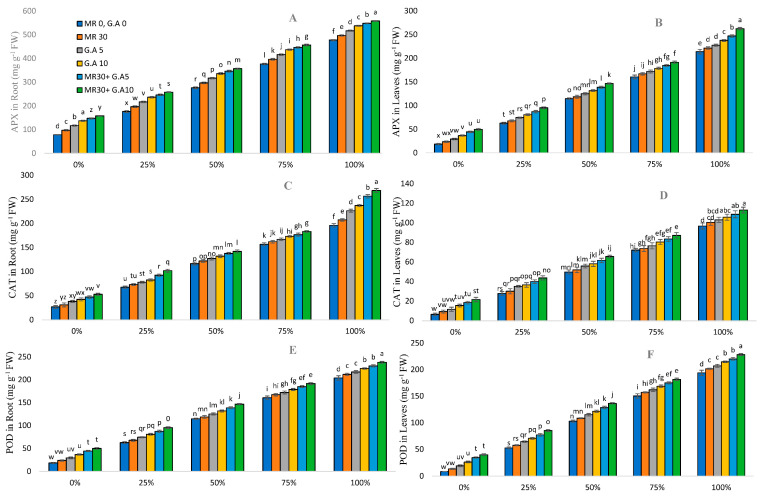
Surgical and textile effluent and glutamic acid effect on APX in roots (**A**), APX in leaves (**B**), CAT in roots (**C**), CAT in leaves (**D**), POD in roots (**E**), POD in leaves (**F**), SOD in roots (**G**), and SOD in leaves (**H**) in *Sorghum* L. Values are demonstrated as means of three replicates along with standard deviation. Different small letters indicate that values are significantly different at *p* < 0.05.

**Table 1 molecules-27-04004-t001:** Effect of surgical and textile effluent alone and/or in combination with GA on heavy metals uptake (µg g^−1^) and accumulation (µg plant^−1^) in microwave-treated and -untreated sorghum.

Wastewater Conc.	Cd Concentration (µg g^−1^)					Cd Accumulation (µg Plant^−1^)				
	0%	25%	50%	75%	100%	0%	25%	50%	75%	100%
Treatments	Leaf					Leaf				
MR 0, G.A 0	0.00 ± 0.00 u	136.81 ± 15.04 t	438.33 ± 20.20 o	757.67 ± 15.01 i	1035 ± 8 d	0.04 ± 0.02 q	1256.29 ± 170.59 p	3203 ± 255.37 mn	3838.8 ± 113.00 lm	3209.33 ± 128.30 mn
MR 30	0.00 ± 0.00 u	171.66 ± 12.58 t	497.66 ± 13.01 n	812.67 ± 15.04 h	1066 ± 7.94 cd	0.07 ± 0.04 q	1806.66 ± 218.30 op	4065.1 ± 163.57 klm	5121.8 ± 256.95 hij	4442.36 ± 192.21 jkl
G.A 5	0.03 ± 0.02 u	216.66 ± 15.27 s	537.33 ± 12.50 m	869.33 ± 18.00 g	1092.33 ± 7.02 c	0.48 ± 0.30 q	2566.66 ± 238.60 no	4872.66 ± 170.98 ijk	6320.16 ± 348.77 efg	5680.6 ± 145.62 ghi
G.A 10	0.05 ± 0.01 u	266.67 ± 12.58 r	583 ± 9.16 l	917.33 ± 10.50 f	1145.34 ± 15.01 b	0.84 ± 0.22 q	3442.83 ± 254.25 lmn	6124.5 ± 385.81 fgh	7462.033 ± 225.17 cd	7064.4 ± 266.35 def
MR 30 + G.A 5	0.08 ± 0.00 u	310.67 ± 16.01 q	631.34 ± 15.04 k	943.34 ± 8.08 f	1195.66 ± 10.06 a	1.38 ± 0.09 q	4447.66 ± 120.60 jkl	7150.33 ± 238.75 de	8808 ± 560.63 b	8768.66 ± 702.66 b
MR 30 + G.A 10	0.09 ± 0.00 u	353.33 ± 13.50 p	673.33 ± 13.50 j	984.33 ± 13.01 e	1231.33 ± 15.04 a	1.71 ± 0.09 q	5413.33 ± 107.50 ghij	8304.66 ± 800.34 bc	10167 ± 449.16 a	10256.33 ± 607.15 a
	Stem					Stem				
MR 0, G.A 0	0.00 ± 0.00 v	160 ± 20 u	623.33 ± 17.56 o	1036.67 ± 20.55 j	1369.33 ± 10.06 e	0.07 ± 0.02 q	2346.66 ± 313.90 p	8207.5 ± 302.37 lm	12482.1 ± 819.41 ij	13512 ± 399.43 ghi
MR 30	0.00 ± 0.00 v	230 ± 25 t	683.67 ± 16.04 n	1080.33 ± 15.50 ij	1415.67 ± 15.04 de	0.11 ± 0.03 q	3825 ± 306.47 op	9685.5 ± 309.73 kl	14399.33 ± 454.31 gh	16515.67 ± 824.14 ef
G.A 5	0.02 ± 0.01 v	297.66 ± 20.40 s	743 ± 18.08 m	1134.67 ± 20.40 hi	1467.66 ± 11.71 cd	0.39 ± 0.18 q	5402 ± 324.22 no	11070.5 ± 288.99 jk	17251.4 ± 697.76 e	19372.8 ± 1054.41 d
G.A 10	0.05 ± 0.01 v	361.67 ± 20.21 r	783.66 ± 13.05 m	1186 ± 6.55 gh	1523.33 ± 21.73 bc	1.16 ± 0.29 q	6932.5 ± 412.39 mn	12695.67 ± 357.21 hij	19610.67 ± 702.446 d	21826.67 ± 593.78 c
MR 30 + G.A 5	0.07 ± 0.01 v	433.33 ± 30.13 q	845.33 ± 17.89 l	1235.33 ± 11.06 fg	1573 ± 11.78 b	1.17 ± 0.30 q	8879.16 ± 567.79 l	14799.17 ± 733.41 fg	22651.67 ± 905.45 c	24851.4 ± 429.32 b
MR 30 + G.A 10	0.09 ± 0.01 v	524.66 ± 30.27 p	920.66 ± 22.72 k	1277.33 ± 9.29 f	1650 ± 32.78 a	2.20 ± 0.08 q	11473.33 ± 692.30 jk	17070.4 ± 877.58 e	25118.33 ± 622.82 b	28598.33 ± 1035.78 a
	Root					Root				
MR 0, G.A 0	0.02 ± 0.01 r	293.76 ± 11.93 q	910 ± 20 m	1645 ± 15 h	2055 ± 18.02 de	0.169 ± 0.08 m	2146.49 ± 166.45 l	5613 ± 237 ij	6911 ± 392.00 fghi	6506.16 ± 181.94 hi
MR 30	0.03 ± 0.01 r	380 ± 20 pq	1000 ± 18.02 m	1731.67 ± 30.14 h	2136.67 ± 25.16 cd	0.281 ± 0.09 m	3118.66 ± 240.01 kl	6735.83 ± 327.96 ghi	8316 ± 490.11 ef	7477.33 ± 175.28 fgh
G.A 5	0.06 ± 0.01 r	463.33 ± 25.16 p	1136.67 ± 50.33 l	1838.33 ± 37.52 g	2210 ± 30 bc	0.69 ± 0.17 m	4297.66 ± 347.91 jk	8268 ± 647.70 efg	10053.33 ± 483.45 cd	9058 ± 285.88 de
G.A 10	0.07 ± 0.01 r	566.67 ± 27.53 o	1326.67 ± 25.16 k	1888.33 ± 36.17 fg	2295 ± 35 b	0.89 ± 0.20 m	5593.83 ± 356.00 ij	10529.67 ± 612.51 cd	11327.83 ± 93.53 c	10784.17 ± 405.35 c
MR 30 + G.A 5	0.08 ± 0.01 r	635 ± 18.02 no	1436 ± 67 ± 45.09 j	1965 ± 25 ef	2398.33 ± 50.08 a	1.02 ± 0.13 m	6766.66 ± 195.53 ghi	13467.33 ± 924.91 b	13624 ± 286.54 b	13664 ± 466.26 b
MR 30 + G.A 10	0.09 ± 0.01 r	725 ± 40.92 n	1538.33 ± 25.65 i	1991.67 ± 24.66 e	2473.34 ± 76.53 a	1.23 ± 0.18 m	8275.16 ± 103.39 efg	16144.17 ± 509.53 a	15735.83 ± 484.71 a	16100.83 ± 1724.46 a

Values are demonstrated as means of three replicates along with standard deviation. Different small letters indicate that values are significantly different at *p* < 0.05.

**Table 2 molecules-27-04004-t002:** Effect of surgical and textile effluent alone and/or in combination with GA on heavy metals uptake (µg g^−1^) and accumulation (µg plant^−1^) in microwave-treated and -untreated sorghum.

Wastewater Conc.	Cu Concentration (µg g^−1^)					Cu Accumulation (µg Plant^−1^)				
	0%	25%	50%	75%	100%	0%	25%	50%	75%	100%
Treatments	Leaf					Leaf				
MR 0, G.A 0	0.00 ± 0.00 t	107.66 ± 11.23 s	259.67 ± 5.50 n	407.66 ± 12.85 i	634 ± 9.54 e	0.03 ± 0.02 o	985.33 ± 82.28 n	1895.33 ± 70.99 jk	2064.53 ± 20.70 klm	1965.433 ± 71.86 klm
MR 30	0.00 ± 0.00 t	133.67 ± 7.02 rs	286.33 ± 12.22 mn	446 ± 19.52 h	660 ± 15.39 e	0.07 ± 0.02 o	1402.16 ± 65.10 mn	2337.76 ± 86.62 jkl	2808.6 ± 113.45 ghij	2749.26 ± 91.84 ij
G.A 5	0.03 ± 0.00 t	152.66 ± 11.06 qr	310.66 ± 8.50 lm	478.33 ± 6.43 gh	716.66 ± 18.50 d	0.48 ± 0.06 o	1804.83 ± 98.61 lm	2816.93 ± 95.17 ghij	3475.6 ± 117.76 efg	3726.06 ± 87.43 ef
G.A 10	0.05 ± 0.00 t	173 ± 8 pq	326.67 ± 5.86 kl	500.34 ± 11.67 g	780.33 ± 13.01 c	0.84 ± 0.07 o	2230.36 ± 74.67 jkl	3429.66 ± 165.86 efgh	4068.8 ± 87.67 de	4811.23 ± 94.46 c
MR 30 + G.A 5	0.07 ± 0.01 t	194.34 ± 10.26 op	346 ± 7.93 jk	536.66 ± 12.09 f	829.66 ± 13.01 b	1.16 ± 0.10 o	2788.33 ± 242.92 hij	3923.33 ± 268.61 ef	5004.33 ± 196.72 c	6080 ± 397.93 b
MR 30 + G.A 10	0.08 ± 0.01 t	218.66 ± 1.52 o	372.66 ± 7.02 j	564.33 ± 10.26 f	881.67 ± 15.01 a	1.46 ± 0.18 o	3353.33 ± 145.11 fghi	4600.66 ± 500.38 cd	5834.33 ± 411.86 b	7352.33 ± 626.62 a
	Stem					Stem				
MR 0, G.A 0	0.00 ± 0.00 t	127.33 ± 11.67 s	283 ± 8.18 n	431 ± 17.08 i	656.33 ± 12.22 e	0.04 ± 0.04 p	1868.33 ± 200.64 o	3726.5 ± 147.63 lm	5184.1 ± 252.73 ij	6474.4 ± 101.00 g
MR 30	0.00 ± 0.00 t	153.73 ± 7.12 rs	305 ± 10.58 mn	466.33 ± 19.00 h	681.66 ± 13.31 e	0.11 ± 0.03 p	2562.46 ± 156.19 no	4321.5 ± 196.36 kl	6211.33 ± 123.78 gh	7957.66 ± 539.62 ef
G.A 5	0.05 ± 0.01 t	172.66 ± 11.06 qr	331 ± 8.71 lm	498.34 ± 6.42 g	733 ± 14.42 d	1.05 ± 0.31 p	3142.16 ± 329.86 mn	4931.5 ± 120.53 ijk	7575.2 ± 226.81 f	9668.66 ± 334.23 d
G.A 10	0.07 ± 0.01 t	193 ± 8 pq	347.66 ± 5.77 kl	521 ± 11.13 g	797 ± 8.71 c	1.44 ± 0.21 p	3697.83 ± 109.06 lm	5630.86 ± 25.74 hi	8611.2 ± 184.82 e	11425.67 ± 545.23 c
MR 30 + G.A 5	0.08 ± 0.01 t	214.33 ± 10.26 op	365 ± 8.18 jk	557.66 ± 11.37 f	850.33 ± 13.01 b	1.81 ± 0.23 p	4397.16 ± 314.65 jkl	6386.66 ± 195.67 gh	10225 ± 424.81 d	13435.13 ± 353.28 b
MR 30 + G.A 10	0.08 ± 0.00 t	235.33 ± 5.68 o	392.66 ± 7.02 j	584.33 ± 10.26 f	901.66 ± 15.01 a	2.05 ± 0.12 p	5146.8 ± 176.66 ij	7275.86 ± 149.42 f	11491 ± 354.51 c	15624 ± 325.91 a
	Root					Root				
MR 0, G.A 0	0.00 ± 0.00 u	194.51 ± 7.16 t	487.66 ± 13.65 o	763.33 ± 17.55 j	1029.33 ± 24.00 f	0.03 ± 0.03 p	1419.61 ± 70.07 o	3006.53 ± 77.40l m	3203.66 ± 79.41 klm	3257.8 ± 50.75 klm
MR 30	0.03 ± 0.00 u	242.66 ± 8.02 s	531.33 ± 12.50 no	790.34 ± 20.00 ij	1139.33 ± 14.36 e	0.06 ± 0.02 p	1989.26 ± 55.33 no	3576.96 ± 111.17 jkl	3792.33 ± 141.66 ijkl	3986.933 ± 81.83 ijk
G.A 5	0.05 ± 0.01 u	291.77 ± 7.12 r	565.34 ± 14.50 mn	834.67 ± 19.29 hi	1240.34 ± 3.05 d	0.51 ± 0.11 p	2703.99 ± 109.99 mn	4105.66 ± 39.75 hij	4562.033 ± 128.88 fghi	5085.1 ± 205.76 f
G.A 10	0.06 ± 0.01 u	342.33 ± 16.62 q	610 ± 13.52l m	877.33 ± 16.01 h	1344.33 ± 21.36 c	0.73 ± 0.14 p	3376 ± 111.33 ljklm	4839.06 ± 214.35 fgh	5263.5 ± 95.00 ef	6318.83 ± 304.45 d
MR 30 + G.A 5	0.07 ± 0.00 u	385 ± 13.45 pq	645.33 ± 18.17 l	930.66 ± 18.00 g	1439.67 ± 15.53 b	0.93 ± 0.89 p	4110.33 ± 337.3 ghij	6043.23 ± 237.33 de	6454 ± 230.78 d	8205.9 ± 382.89 b
MR 30 + G.A 10	0.08 ± 0.00 u	428.66 ± 16.92 p	698.33 ± 8.32 k	958.67 ± 21.22 g	1535 ± 21.79 a	1.14 ± 0.11 p	4904 ± 362.67 fg	7333.16 ± 382.17 c	7575.83 ± 328.87 bc	9984.16 ± 896.42 a

Values are demonstrated as means of three replicates along with standard deviation. Different small letters indicate that values are significantly different at *p* < 0.05.

**Table 3 molecules-27-04004-t003:** Effect of surgical and textile effluent alone and/or in combination with GA on heavy metals uptake (µg g^−1^) and accumulation (µg plant^−1^) in microwave-treated and -untreated sorghum.

Wastewater Conc.	Pb Concentration (µg g^−1^)					Pb Accumulation (µg Plant^−1^)				
	0%	25%	50%	75%	100%	0%	25%	50%	75%	100%
Treatments	Leaf					Leaf				
MR 0, G.A 0	0.00 ± 0.00 t	135.33 ± 10.40 s	316.66 ± 28.43 no	445.67 ± 12.05 jk	656.33 ± 12.22 e	0.03 ± 0.02 p	1239.16 ± 73.18 o	2308.83 ± 175.43 lmn	2258.13 ± 83.02 lmn	2034.36 ± 64.57 mno
MR 30	0.01 ± 0.00 t	169 ± 13.11 rs	344 ± 17.69 mn	466.33 ± 19.00 ij	761.66 ± 17.56 d	0.13 ± 0.10 p	1771.83 ± 106.86 no	2808.83 ± 124.15 jklm	2936.7 ± 111.31 jkl	3172.33 ± 82.97 jk
G.A 5	0.04 ± 0.01 t	186 ± 11.53 rs	367.66 ± 19.00l mn	501.67 ± 11.01 hi	816 ± 26.51 c	0.57 ± 0.13 p	2199.16 ± 96.87 lmn	3332.26 ± 138.43 ijk	3644.26 ± 98.12 hij	4242.03 ± 102.89 gh
G.A 10	0.05 ± 0.02 t	213 ± 21.07 qr	390.33 ± 3.05 lm	527.66 ± 22.30 gh	954.66 ± 27.97 b	0.84 ± 0.32 p	2742.7 ± 194.95 klm	4098.16 ± 187.91 ghi	4290.26 ± 143.24 fgh	5886.033 ± 180.33 cd
MR 30 + G.A 5	0.08 ± 0.01 t	253 ± 7.55 pq	409 ± 10.58 kl	561.33 ± 16.50 fg	992.33 ± 14.36 b	1.34 ± 0.25 p	3628.66 ± 243.28 hij	4638 ± 328.95 efg	5233.66 ± 202.36 cde	7271.66 ± 464.80 b
MR 30 + G.A 10	0.09 ± 0.00 t	282.33 ± 16.16 op	415.66 ± 16.16 jkl	589.34 ± 15.17 f	1048.34 ± 36.01 a	1.58 ± 0.05 p	4334 ± 391.84 fgh	5128.33 ± 547.92 def	6094.33 ± 476.07 c	8748.33 ± 890 a
	Stem					Stem				
MR 0, G.A 0	0.00 ± 0.00 v	116.66 ± 15.27 u	418.34 ± 20.21 p	721.66 ± 17.03 k	1000.67 ± 3.05 ef	0.05 ± 0.03 r	1710 ± 226.49 q	5507.5 ± 272.70 mn	8690.26 ± 601.52 ijk	9873.6 ± 254.35 hi
MR 30	0.03 ± 0.02 v	151.67 ± 12.58 tu	477.67 ± 13.01 o	778 ± 14.52 jk	1031 ± 7.93 de	0.58 ± 0.47 r	2523.33 ± 132.03 pq	6766.83 ± 223.43 lm	10373.67 ± 498.06 h	12029.33 ± 631.14 g
G.A 5	0.05 ± 0.01 v	196.66 ± 15.27 st	513.66 ± 7.67 no	834.66 ± 18.50 ij	1061 ± 10.14 cd	0.98 ± 0.20 r	3566.66 ± 189.03 op	7653 ± 82.86 jkl	12690.6 ± 552.34 fg	14007.33 ± 830.28 ef
G.A 10	0.06 ± 0.01 v	246.66 ± 12.58 rs	564.33 ± 14.01 mn	884.66 ± 7.63 hi	1110 ± 15 bc	1.24 ± 0.21 r	4727.5 ± 241.41 no	9142.33 ± 303.61 hij	14628.93 ± 567.05 de	15905 ± 462.41 cd
MR 30 + G.A 5	0.07 ± 0.00 v	257.33 ± 44.23 r	578.33 ± 60.92 m	908.66 ± 8.08 gh	1160 ± 10 ab	1.74 ± 0.17 r	5272.83 ± 891.51 mn	10125.67 ± 1181.01 hi	16661.33 ± 649.43 c	18326 ± 278.47 b
MR 30 + G.A 10	0.09 ± 0.00 v	333.33 ± 13.50 q	653 ± 13 l	950 ± 13.51 fg	1199.33 ± 12.09 a	2.20 ± 0.08 r	7288.93 ± 308.44 kl	12106.6 ± 568.01 g	18679 ± 363.23 b	20787 ± 655.94 a
	Root					Root				
MR 0, G.A 0	0.00 ± 0.00 v	141.66 ± 15.27 u	443.33 ± 20.21 p	763 ± 15 j	1040 ± 8 e	0.03 ± 0.01 q	1037.16 ± 152.62 p	2735 ± 169.55 lmn	3206.6 ± 215.60 klm	3293.33 ± 112.62 jklm
MR 30	0.04 ± 0.03 v	176.67 ± 12.58 u	503 ± 13 o	818 ± 14.52 i	1071 ± 7.93 de	0.23 ± 0.29 q	1450.33 ± 138.63 op	3388.6 ± 189.70 jkl	3928.33 ± 233.11 hij	3748.1 ± 88.22 ijk
G.A 5	0.06 ± 0.01 v	221.66 ± 15.27 t	542.33 ± 12.50 n	874.67 ± 18.50 h	1101 ± 10.14 d	0.44 ± 0.05 q	2056.66 ± 196.21 no	3943.03 ± 226.55 hij	4783.36 ± 233.36 ef	4515.2 ± 229.27 fgh
G.A 10	0.06 ± 0.01 v	271.67 ± 12.58 s	588 ± 9.16 m	924.33 ± 8.08 g	1150.33 ± 15.01 c	0.69 ± 0.12 q	2681.66 ± 164.17 mn	4666.46 ± 254.30 fg	5546.53 ± 140.69 d	5408.56 ± 300.50 de
MR 30 + G.A 5	0.08 ± 0.00 v	315.66 ± 16.01 r	636.33 ± 15.04 l	948.67 ± 8.08 g	1200 ± 10 b	1.06 ± 0.04 q	3367.33 ± 257.68 jklm	5963.8 ± 363.22 cd	6577.33 ± 116.02 bc	6839.66 ± 312.09 b
MR 30 + G.A 10	0.09 ± 0.00 v	358.34 ± 13.50 q	678.33 ± 13.50 k	990 ± 13.52 f	1238 ± 13.31 a	1.27 ± 0.11 q	4092.46 ± 47.30 ghi	7118 ± 197.50 b	7820.9 ± 196.96 a	8047.833 ± 595.41 a

Values are demonstrated as means of three replicates along with standard deviation. Different small letters indicate that values are significantly different at *p* < 0.05.

## Data Availability

Not applicable.

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
