# Peer review of "Microwave Irradiation and Glutamic Acid-Assisted Phytotreatment of Textile and Surgical Industrial Wastewater by Sorghum"

_molecules, 2022, doi:10.3390/molecules27134004_

Round 1

Reviewer 1 Report

The artile is concerned with treatment of industrial wastewater by microwave irradiation. The influence of doses of microwave irradiation on seed germination in Sorghum was investigated. The article is worth publishing before some questions are answered. 

(1) the irradiation condition and test methods need more description.

(2) the notation of figures 1-4 is not clear. 

(3) there are many grammar problems. 

(4) some references  do not have pages. 

Author Response

Comments and Suggestions for Authors

The article is concerned with treatment of industrial wastewater by microwave irradiation. The influence of doses of microwave irradiation on seed germination in Sorghum was investigated. The article is worth publishing before some questions are answered.

Response:

The authors are thankful to the reviewer for their valuable comments/suggestions to improve our manuscript along with the validation of our research work. The authors have addressed all the suggestions in revised manuscript

  • the irradiation condition and test methods need more description.

Response:

The required information has been added in the material and method section.

  • the notation of figures 1-4 is not clear.

Response:

The figure notations have been readjusted. I think it is not properly readable due to the page margin, small page size etc. however, the notations have been rechecked and readjusted in revised MS for clear display to the readers.

  • there are many grammar problems.

Response:

The whole article has been thoroughly read to remove the all-grammar issues.

  • some references do not have pages.

Response:

The missing page numbers of the references have been added in the revised MS.

Reviewer 2 Report

Reviewer’s Comments:

The concentration percentage is not clear. What units you use?? The increase percentage is also not clear (mass, volumetric, i.e.). Improved them and included clearly the units employed in Materials and Methods Section (4).

In the tables (1, 2 and 3), include the time unit used.

Do not use so many abbreviations in Conclusions.

The results are interested, but the compression is not good due to the graphs employed. Should found other forms to display their results. Also, the tables could be improved. The tables are excessively extensive and the importance of the data displayed is lost. I suggested that changed some figures, to improved them. Don't just use bar graphics, is not adequate.

SF 1 and 2 (Material Supplementary), Figure 2 the scale of x axes is cut and the last legend "green bar" is not displayed.

Author Response

Reviewer’s Comments:

The concentration percentage is not clear. What units you use?? The increase percentage is also not clear (mass, volumetric, i.e.). Improved them and included clearly the units employed in Materials and Methods Section (4).

Response:

The authors are thankful to the reviewer for their valuable comments/suggestions to improve our manuscript.

The calculations are made to express the relative percentage increase and decrease in the manuscript. As there is no unit to express the percentage increase and decrease rather the unit of the parameter being discussed. We can say that, “the activities of SOD antioxidant enzyme were increase by 87% as compared to the respective control under 50% wastewater treatment”. In this case the unit will be the same as the unit of antioxidant enzyme SOD. However, the result section has been read to make clarity and better understanding for the readers and the same has been added in the material and methods section.

In the tables (1, 2 and 3), include the time unit used.

Response:

The unit of heavy metal concentration and accumulation has been provided in the table caption in revised MS. However, it has already been mentioned in the tables adjacent to the treatments. The tables have also been revised to make clarity for readers.

Do not use so many abbreviations in Conclusions.

Response:

The suggestion has been incorporated in the revised MS.

The results are interested, but the compression is not good due to the graphs employed. Should found other forms to display their results. Also, the tables could be improved. The tables are excessively extensive and the importance of the data displayed is lost. I suggested that changed some figures, to improved them. Don't just use bar graphics, is not adequate.

Response:

Dear reviewer, I can understand the difficulty you faced during the review because of the separate figures/ table’s pages through the whole document. It was also difficult for us to insert the tables and figures right after their description. I hope the published version will be of good quality to be read easily. For the convenience of readers, we have tried our best to improve the quality.

Dear reviewer, the bar graph is more suitable for these kind of results as if we use the line graphs then it can be more complicated due to the S.D and lettering clarifying the significance of results. I have considered your suggestion and will incorporate in the future studies. 

SF 1 and 2 (Material Supplementary), Figure 2 the scale of x axes is cut and the last legend "green bar" is not displayed.

Response:

The figures have been checked and readjusted.